**Data Availability Statement:** All relevant data are within the manuscript and Supporting Information files.

**Funding:** This study was funded by the Brain Aging Study from the L.K. Whittier Foundation (to MGH),

# Retinal nerve fiber layer thickness predicts CSF amyloid/tau before cognitive decline

Samuel Asanad[1,2], Michele Fantini[1,2,3], William Sultan[1], Marco Nassisi[1,4], Christian M. Felix[2], Jessica Wu[2], Rustum Karanjia[1,2,5,6], Fred N. Ross-Cisneros[1], Abhay P. Sagare[7], Berislav V. Zlokovic[7], Helena C. Chui[8], Janice M. Pogoda[9], Xianghong Arakaki[10], Alfred N. Fonteh[10], Alfredo A. Sadun A. A.[1,2‡], Michael G. Harrington[10]*‡

**1** Doheny Eye Institute, Los Angeles, CA, United States of America, **2** Department of Ophthalmology, David Geffen School of Medicine at UCLA, Los Angeles, CA, United States of America, **3** Department of Medicine, Ophthalmology, University of Udine, Udine, Italy, **4** Department of Clinical Sciences and Community Health, Ophthalmological Unit, IRCCS-Cà Granda Foundation—Ospedale Maggiore Policlinico, University of Milan, Milan, Italy, **5** Department of Ophthalmology, University of Ottawa, Ottawa, Ontario, Canada, **6** Ottawa Hospital Research Institute, Ottawa, Ontario, Canada, **7** Department of Physiology and Neuroscience, Zilkha Neurogenetic Institute, Keck School of Medicine, University of Southern California, Los Angeles, CA, United States of America, **8** Department of Neurology, University of Southern California, Los Angeles, CA, United States of America, **9** Cipher Biostatistics & Reporting, Reno, NV, United States of America, **10** Huntington Medical Research Institutes, Pasadena, CA, United States of America

‡ These authors are joint senior authors on this work.

* michael.harrington@hmri.org

## Abstract

### Background

Alzheimer's disease (AD) pathology precedes symptoms and its detection can identify at-risk individuals who may benefit from early treatment. Since the retinal nerve fiber layer (RNFL) is depleted in established AD, we tested whether its thickness can predict whether cognitively healthy (CH) individuals have a normal or pathological cerebrospinal fluid (CSF) $A\beta_{42}$ (A) and tau (T) ratio.

### Methods

As part of an ongoing longitudinal study, we enrolled CH individuals, excluding those with cognitive impairment and significant ocular pathology. We classified the CH group into two sub-groups, normal (CH-NAT, n = 16) or pathological (CH-PAT, n = 27), using a logistic regression model from the CSF AT ratio that identified >85% of patients with a clinically probable AD diagnosis. Spectral-domain optical coherence tomography (OCT) was acquired for RNFL, ganglion cell-inner plexiform layer (GC-IPL), and macular thickness. Group differences were tested using mixed model repeated measures and a classification model derived using multiple logistic regression.

### Results

Mean age (± standard deviation) in the CH-PAT group (n = 27; 75.2 ± 8.4 years) was similar (p = 0.50) to the CH-NAT group (n = 16; 74.1 ± 7.9 years). Mean RNFL (standard error) was

National Institute of Aging P01 AG052350 (to BVZ) and National Institute of Aging P50 AG005142 (to HCC). The funders had no role in study design, data collection and analysis, decision to publish, or preparation of the manuscript.

**Competing interests:** The authors have declared that no competing interests exist.

thinner in the CH-PAT group by 9.8 (2.7) µm; p < 0.001. RNFL thickness classified CH-NAT vs. CH-PAT with 87% sensitivity and 56.3% specificity.

## Conclusions

Our retinal data predict which individuals have CSF biomarkers of AD pathology before cognitive deficits are detectable with 87% sensitivity. Such results from easy-to-acquire, objective and non-invasive measurements of the RNFL merit further study of OCT technology to monitor or screen for early AD pathology.

## Introduction

Alzheimer's disease (AD) is the leading cause of dementia and one of the most expensive diseases in America at a total cost of $290 billion in 2019 [1–3]. Over 26 million people are currently affected by AD worldwide, however this prevalence is estimated to quadruple by 2050. Widespread screening approaches are urgently needed to diagnose early disease stages to allow therapy trials before symptomatic decline [4,5].

In considering accessible and noninvasive screening approaches, ophthalmologic impairments including contrast sensitivity, color recognition, and motion perception have been reported in early stages of dementia, even prior to manifesting as a clear diagnosis of AD [6–12]. These visual dysfunctions in AD have been associated with degeneration of the optic nerve and retina, developmental outgrowths of the brain [13–15], since cortical deficits alone do not explain these visual deficits [11,12,16,17]. The first ocular manifestations of AD were shown by our laboratory, where the optic nerves of postmortem tissue exhibited diffuse axonal degeneration [18,19]. More recently, hallmark pathologies of neurodegeneration typical for AD in the brain have also been suggested in the retina. We previously demonstrated amyloid β-protein (Aβ) accumulation inside and around melanopsin-containing RGCs (mRGCs) in AD, possibly explaining circadian rhythm disturbances seen in these patients [20]. Additional studies have identified Aβ plaque pathology in postmortem retinas of AD patients as well as in early-stage disease [1,20–24]. Following the advent of optical coherence tomography (OCT), *in vivo* studies have revealed significant retinal thinning [16, 25–28]; however, the clinical utility of the retina as a biomarker for early AD has not been established [29–34].

The pre-symptomatic phase of AD is recognized in cognitively healthy (CH) individuals having AD pathology many years prior to symptom onset, based on altered amyloid and tau biomarkers detected using positron emission tomography (PET) or cerebrospinal fluid (CSF) analysis [35,36]. For example, our laboratory recently demonstrated that pre-symptomatic AD can be detected more sensitively and specifically by the CSF $A\beta_{42}$/Tau ratio relative to the concentration of these biomarkers independently [37, 38]. The purpose of this study was to investigate whether retinal thickness can predict whether CH individuals have a normal or pathological $A\beta_{42}$/Tau ratio.

## Materials & methods

### Human participants

The Institutional Review Boards of both the Huntington Medical Research Institutes (HMRI), Pasadena (IRB #33797) and the University of California Los Angeles (IRB #17001645) approved the protocol and consent forms for this study, which was conducted and performed

in compliance with the ethical standards set out in the Declaration of Helsinki. Prior to enrollment, all study participants gave written, informed consent. Clinical and CSF studies were performed at HMRI, and ophthalmology assessments were conducted at the Doheny Eye Center, Department of Neuro-ophthalmology, Pasadena, CA.

Participants over 60 years of age were recruited locally at HMRI for aging research if they had no cognitive impairment after medical and neuropsychological assessment, using the Uniform Data Set criteria of the National Alzheimer's Coordinating Center (https://www.alz.washington.edu/WEB/npsych_means.html) and after consensus clinical conference. CSF $Aß_{42}$/Tau ratios were determined from lumbar fluid as described [37]. MSD assay (catalog no. K15199G, MSD, Rockville, MD) was used to determine CSF levels of $Aβ_{42}$. MSD assay (catalog no. K15121G, MSD, Rockville, MD) was used to determine CSF levels of total tau. We used a logistic regression cutoff for the CSF $Aß_{42}$/Tau ratio that correctly classified more than 85% of probable AD patients to identify two separate cohorts of the CH participants: those with Normal $Aß_{42}$/Tau ratio ("CH-NAT"), or those with Pathological $Aß_{42}$/Tau ratio ("CH-PAT"). We thus defined that CH-PAT individuals had preclinical AD.

Ophthalmic examination included tonometry, assessment of best-corrected visual acuity (BCVA), slit lamp examination of the anterior segment, and dilated fundus examination for all participants. Exclusion criteria for all groups were: BCVA < 20/50; refractive error $> \pm 5$ spherical equivalent; poor image quality defined by an image signal less than 6 due to severe cataracts or unstable fixation; intraocular pressure (IOP) > 20mm Hg; pre-existing macular pathologies such as age-related macular degeneration, epiretinal membrane or macular hole; other retinopathies such as retinal vascular occlusion or retinal dystrophy; pre-existing ocular diseases such as glaucoma, optic neuropathy or uveitis; previous intraocular surgery or laser treatment except for cataract surgery performed at least 12 months prior to enrolment; penetrating ocular trauma; current active smoking status; and history or evidence of other neurological or psychiatric disorders, diabetes mellitus, poorly controlled systemic arterial hypertension defined by systolic > 150 and diastolic > 90, cardiovascular diseases, renal failure, substance abuse in the past 5 years, systemic corticosteroid use exceeding 6 months, and systemic autoimmune conditions with associated optic neuropathies.

## Instrumentation and procedures

RNFL, RGC-IPL, and full macular thicknesses were measured for all participants using spectral-domain OCT (Cirrus SD-OCT, software v 6.0; Carl Zeiss Meditec). Scans were all acquired using the protocols for Optic Disc Cube 200 x 200 and the Macular Cube 512 x 128 in both eyes with pupil dilation. Following proper seating and alignment of each individual, the iris was brought into view using the mouse-driven alignment system, and the ophthalmoscopic image was focused. To acquire the Optic Disc Cube as seen in Fig 1A, the optic nerve head was centered on the live image, and centering and enhancement were optimized. After launching the scanning process, the instrument's 840 nm wavelength laser beam generated a cube of data measuring 6 mm x 6 mm after scanning a series of 200 B-scans with 200 A-scans per B-scan (40,000 points) in 1.5 seconds (27,000 A-scans/sec). Cirrus SD-OCT algorithms were used to find the optic disc with automatic placement of a calculation circle measuring 3.46 mm in diameter symmetrically around it. Layerseeking algorithms were used to find the RNFL inner (anterior) boundary and the RNFL outer (posterior) boundary for the entire cube except the optic disc. The system extracted 256 A-scan samples from the data cube along the path of the calculation circle.

With respect to acquiring the Macular Cube, participants fixated on the central target. The ganglion cell analysis algorithm detected and measured the thickness of the macular RGC-IPL

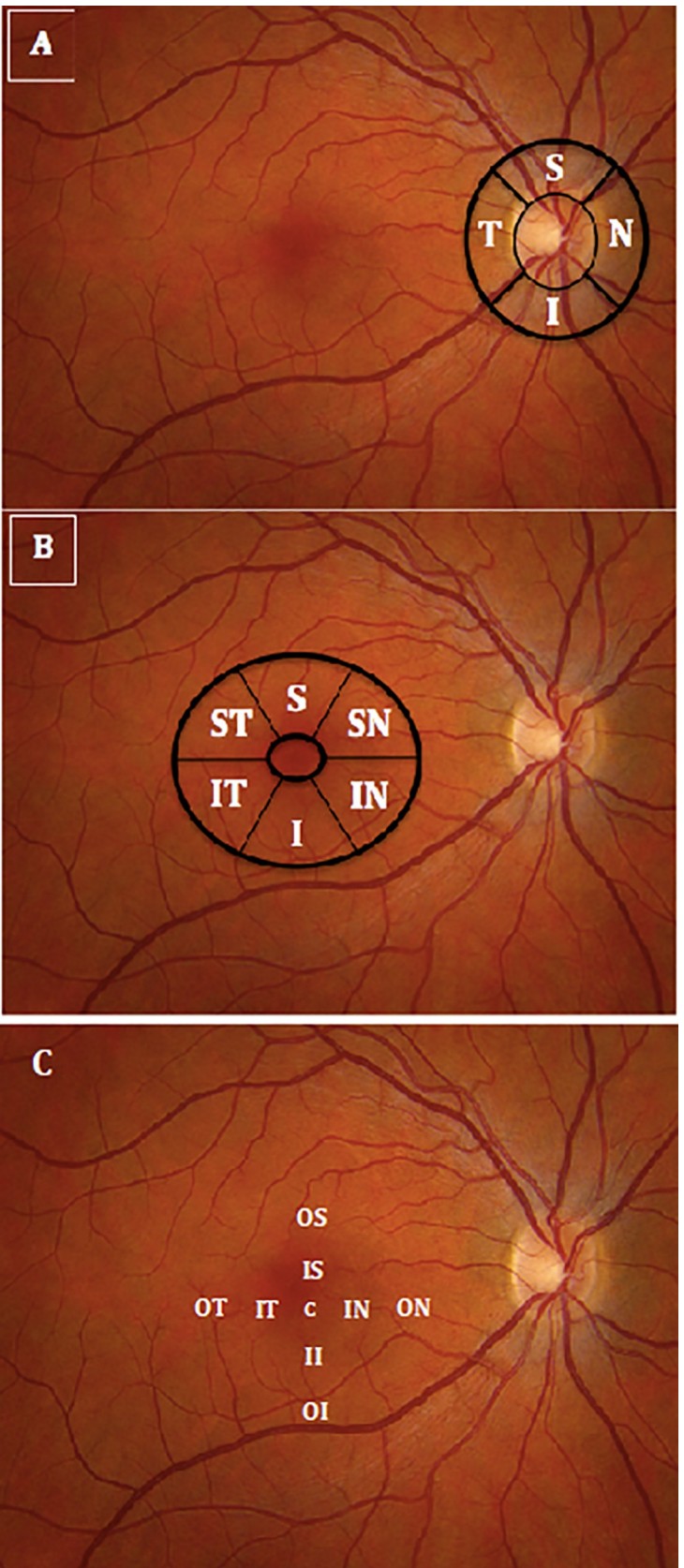

**Fig 1. Illustrates optical coherence tomography imaging of the peripapillary and macular areas.** (A) The thicknesses of the 4 retinal nerve fiber layer quadrants (temporal, superior, nasal, inferior); (B) ganglion cell-inner plexiform layer sectors (superior-temporal, superior, superior-nasal, inferior-nasal, inferior, inferior-temporal; (C) macular full-thickness sectors (center, inner-superior, outer-superior, inner-inferior, outer-inferior, inner-temporal, outer-temporal, inner-nasal, outer-nasal) were measured using peripapillary and macular circular scans centered on the disc and on the fovea, respectively).

within a 14.13-mm$^2$ elliptical annulus area centered on the fovea (Fig 1B). The ganglion cell analysis algorithm processed data from 3-dimensional volume scans and measured the thickness of the macular RGC-IPL. The average, minimum, and 6 sectoral (superotemporal, superior, superonasal, inferonasal, inferior, and inferotemporal) RGC-IPL thicknesses were measured from centering the elliptical annulus on the fovea (Fig 1B). Retinal thickness maps of the total macular region were acquired using the macular cube scan within a $6 \times 6$-mm$^2$ circular area centered on the fovea. Measurements were averaged over 9-retinal subfields, as defined by the Early Treatment Diabetic Retinopathy Study (ETDRS) (Fig 1C). A published, detailed description of this algorithm and how it operates is available for reference [39]. The built-in SD-OCT eye-tracking system provided reproducible measurements with a coefficient of variation of 0.5% [40].

An experienced operator captured all images. Individual scan volumes were reviewed for segmentation errors. Scans with significant motion artifacts, segmentation errors, or signal strength values less than 6 were excluded from analysis. To maximize the reflective signal, polarization was optimized and the scan with the best centration of the optic disc was consistently selected. The thicknesses of the three regions were then compared between the two cohorts and correlated with their CSF levels of Aß$_{42}$, Tau, and Aß$_{42}$/Tau ratio.

## Statistical methods

Comparisons between groups on retinal thickness were made using mixed model repeated measures with unstructured covariance. Group (CH-NAT, CH-PAT), layer (RNFL, GC-IPL, macula), side (OD, OS), and region (S, N, I, T) were fixed effects. Error degrees of freedom was calculated using the Kenward-Rogers approximation. Model assumptions were checked using Q-Q and scatter plots of residuals. Model-based least-squares (LS) means, standard errors, and p-values for tests of differences in LS means are reported.

Multivariable logistic regression was used to predict CH-PAT group membership (dependent variable) based on retinal thickness (independent variables). Only subjects with all data points available (retinal thickness for all 3 layers, both sides, and all 4 regions) were included. Correlations between pairs of independent variables were assessed to minimize collinearity; for highly correlated pairs ($r > 0.7$), only the variable with the strongest association with group was retained as a candidate predictor. Independently significant predictors constituted the final model. A cutoff value for the probability that a given subject is CH-PAT was selected based on sensitivity and specificity, and the cutoff probability was entered into the left side of the final model equation to derive a decision boundary.

Statistical significance was defined as $p < 0.05$, two-sided. As an exploratory study, no adjustments were made for multiplicity. Analyses were done using SAS v9.4 (SAS Institute, Inc., Cary NC).

## Results

Twenty-seven participants were classified as CH-PAT (mean age ± standard deviation 75.2 ± 8.4 years; 48% female) and 16 as CH-NAT (mean age 74.1 ± 7.4 years; 56% female).

**Table 1. Retinal thickness by layer.**

| Layer | CH-NAT (N = 16 subjects) | | CH-PAT (N = 27 subjects) | | p-value |
|---|---|---|---|---|---|
| RNFL | | | | | |
| n | 128 | | 192 | | |
| Mean (SD) | 88.76 | (24.290) | 84.24 | (24.112) | |
| LS Mean (SE)[1] | 93.27 | (2.201) | 83.46 | (1.812) | 0.0009 |
| Median | 92.75 | | 86.00 | | |
| IQR | 65.00 | - 108.00 | 61.25 | - 102.75 | |
| Range | 48.0 | - 135.0 | 41.0 | - 133.0 | |
| GC-IPL | | | | | |
| n | 186 | | 306 | | |
| Mean (SD) | 74.42 | (6.316) | 73.09 | (6.693) | |
| LS Mean (SE)[1] | 75.06 | (1.312) | 73.13 | (1.068) | 0.22 |
| Median | 74.50 | | 73.50 | | |
| IQR | 70.00 | - 78.50 | 69.00 | - 77.00 | |
| Range | 59.0 | - 93.0 | 52.0 | - 94.0 | |
| Macula | | | | | |
| n | 279 | | 459 | | |
| Mean (SD) | 294.48 | (29.663) | 287.27 | (28.936) | |
| LS Mean (SE)[1] | 291.67 | (4.110) | 288.97 | (3.191) | 0.60 |
| Median | 293.50 | | 290.00 | | |
| IQR | 270.00 | - 318.00 | 264.00 | - 309.50 | |
| Range | 232.5 | - 383.5 | 211.0 | - 355.0 | |

Abbreviations: GC-IPL, ganglion cell-inner plexiform layer; IQR, interquartile range; LS, least-squares; n, number of measurements; RNFL, retinal nerve fiber layer; SD, standard deviation; SE, standard error.

[1] Adjusted for side (OD, OS) and region (S, I, T, N).

There were no significant group differences at the macular or GC-IPL locations in the OCT data (S1 Table). However, mean RNFL was thinner (p = 0.0009) in the CH-PAT than CH-NAT group, as summarized in Table 1. Fig 2 displays the 10μm difference between these groups. The difference between groups in mean RNFL thinning was significant at the nasal location (S2 Table). The CSF Aß$_{42}$/ Tau ratio was more correlated with RNFL thinning compared to individual biomarkers, however examination of the roles of each independent biomarker displayed in the scatter plots of S1 Fig suggests that the CSF total Tau levels correlate more with RNFL thinning than Aß$_{42}$.

Logistic regression was used to predict CH-PAT from S, N, I, T regions for each of OD and OS. The best model included only OD_T and OD_N as predictors (all of the OS measurements were highly correlated with their OD counterparts, but not nearly as predictive). This required omitting from the analysis participants with any missing data of the 8 possible predictors, which resulted in the exclusion of 4 individuals. After determining the best model (OD_T and OD_N), a cutoff for predicted event probability was chosen such that sensitivity was at least 85% and specificity was maximized. The selected cutoff of 0.46 yielded 87% sensitivity and 56% specificity (AUC = 0.83).

## Discussion

These results extend the findings of optic neuropathy being a common feature in AD as first demonstrated by our laboratory over 30 years ago [19]. More recently, the availability of

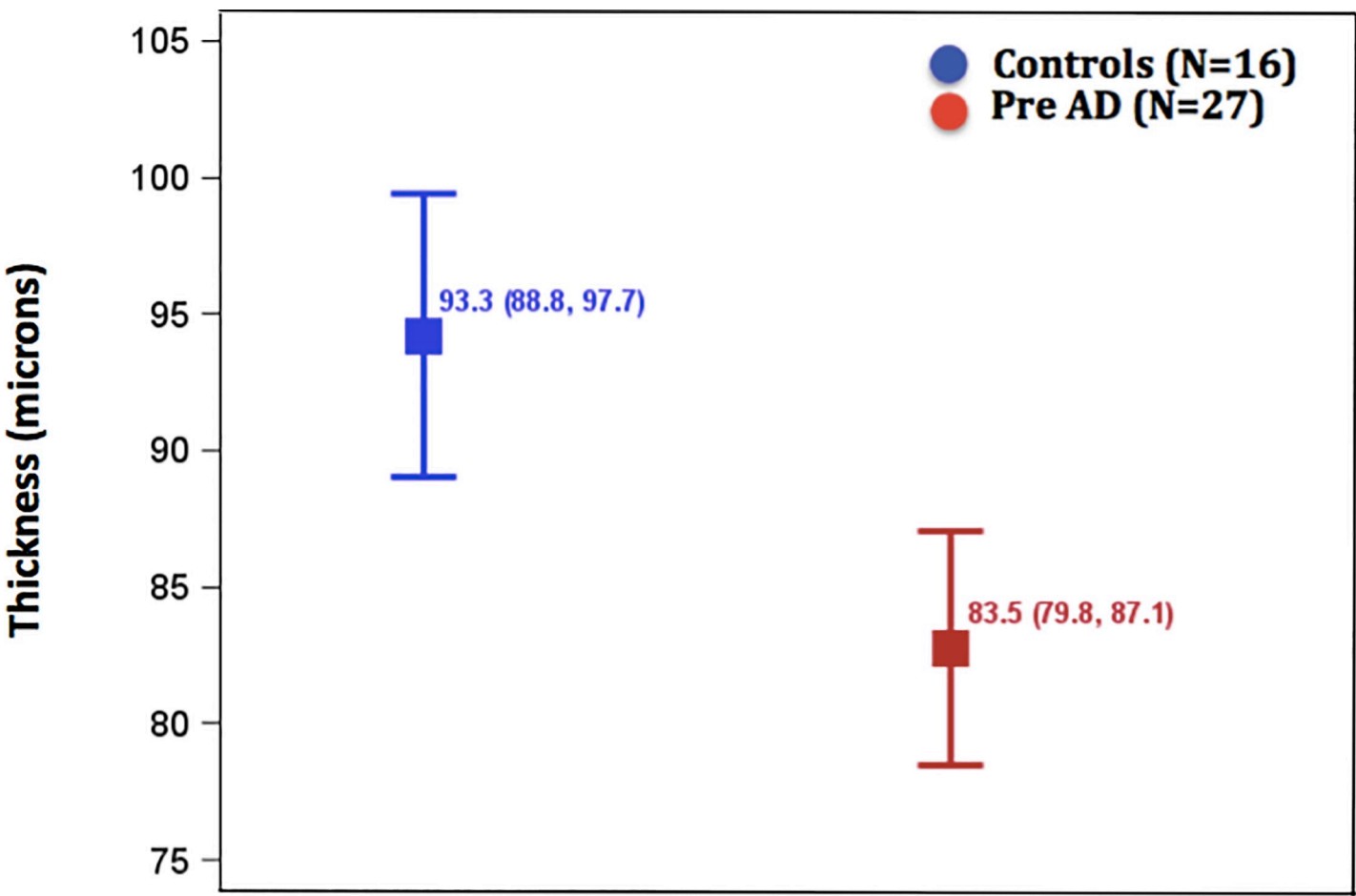

**Fig 2.** Depicts least-squares mean (95% CI) retinal nerve fiber layer (RNFL) thickness adjusted for side and region between cognitively healthy controls (blue) and preclinical AD participants (red).

sensitive and reliable retinal imaging techniques has revealed a number of retinal abnormalities in earlier stages of AD [28]. The major finding from this study is that RNFL thickness measure by OCT was able to predict the biochemical class of CH-NAT vs. CH-PAT in cognitively healthy, older individuals with high sensitivity (Fig 2 and Table 2). This is the first study to predict pre-symptomatic AD pathology with high sensitivity using non-invasive technology and provides an opportunity to encourage widespread screening.

We assessed retinal pathology in relation to both CSF Aß and Tau. When analyzed individually, CSF Tau was more associated with RNFL thickness relative to amyloid. Notably, this observed relationship between CSF protein levels and the retina is similar to that observed when CSF Aß is known to be inversely correlated with Aß load and neuropathology in the brain, whereas CSF Tau is known to be directly correlated. More importantly, the combined

**Table 2. Logistic regression model for the probability of CH-PAT based on retinal thickness in the retinal nerve fiber layer.**

| Parameter | Coefficient | (SE) | Odds Ratio | (95% CI) | p-value |
|---|---|---|---|---|---|
| Intercept | 15.62 | (5.120) | -- | -- | -- |
| OD Side, Region N | -0.10 | (0.041) | 0.91 | (0.84, 0.98) | 0.02 |
| OD Side, Region T | -0.14 | (0.052) | 0.87 | (0.79, 0.97) | 0.008 |

ratio of CSF Aß and Tau has been shown to predict AD neuropathology with high accuracy, outweighing the performance of these proteins individually [41]. Our findings are further supported by recent studies illustrating the role of Tau dysfunction in pre-symptomatic AD, whereby Tau pathology of the retina may precede pathology of the brain and cognitive loss [42]. The present results on the retina support our previous findings in regard to the advantages of using an index for the ratio of two biomarkers for predicting AD pathology [37], suggesting that both amyloid and Tau likely contribute, independently, to retinal pathology.

We found significant RNFL thinning in pre-symptomatic AD participants. Studies in patients with symptomatic cognitive impairment including AD have also shown RNFL thinning, which significantly correlated with visual function and cognitive deficits [43–52]. Studies have also illustrated retinal atrophy of the macular region. Cunha et al. reported thinning of the GC-IPL and macular full-thickness reduction in addition peripapillary RNFL loss [53]. The group found significant correlations between cognitive function testing and retinal thickness in both the peripapillary and macular regions [53]. Additional studies conducted by Martin et al. investigated retinal layer thickness changes in relation to disease duration and severity [25]. Notably, RNFL thinning in AD occurred early while RGCL as well as outer retinal layer thinning occurred later [25]. This progressive pattern of RNFL loss followed by RGCL loss is consistent with previous studies conducted by our laboratory using postmortem histopathology [54]. These tissues were histopathologically confirmed for Aβ deposition and were derived from patients who were neuropathologically confirmed for severe AD. In addition, retinal layer thickness profiles matched the distribution of retinal Aβ deposits in these tissues, as previously demonstrated by our laboratory and others [20, 21,23,54–56].

Our study also showed thinning of the GC-IPL and macular full-thickness regions, though these were not statistically significant. Longitudinal studies conducted by Santos et al. have reported similar findings [57]. Specifically, Santos and colleagues found significant thinning primarily of the RNFL in preclinical AD [57]. Macular thinning involving the inner plexiform and outer nuclear layers were also observed. However, these thickness changes were reportedly attributed to age-related decline, as expected over the timespan of the study. Notably, only the RNFL significantly correlated with neocortical amyloid load as measured by positron emission tomography [57]. These results corroborate our retinal thickness findings and support the premise that nerve fiber layer axonal loss as measured by OCT precedes symptomatic cognitive decline.

Retinal thickness changes have also been investigated in mild cognitive impairment (MCI). Previous OCT studies have illustrated retinal thinning in MCI, although less severe relative to AD [18,30,34,46,50,58–60]. These findings might suggest MCI as a transitional stage between normal cognitive function and dementia. Notably, however, prior studies have been variable in their results. Ascaso et al. and Oktem et al. reported significant RNFL thinning in MCI [34,50], whereas Almeida et al. found significant GC-IPL thinning [61] or no significant retinal thinning [62,63]. In addition, Ascaso et al. and Oktem et al. reported a significant correlation between RNFL thickness and Mini-Mental State Examination (MMSE) scores in MCI [34,50], while Almeida et al. observed a significant correlation between GC-IPL thickness and MMSE scores [61]. In turn, retinal thickness findings in MCI have been controversial and the utility of the retina as a marker of disease conversion from MCI to AD remains inconclusive. These discrepant findings between studies may be attributed to confounding factors including segmentation algorithms, variability between OCT devices, and the diagnostic criteria for AD and MCI [28,64,65]. Researchers have largely relied on the MMSE as a screening measure of general cognitive function. Importantly, however, this test has several reported psychometric limitations and may not always be a reliable cognitive marker [66]. MCI is a complex,

heterogeneous stage of cognitive decline. Patients may not progress to AD, may revert back to normal cognitive health, or may advance to non-AD dementia [67].

Our present study on early AD pathology differed in two ways that are important because the pathology is known to start many years before symptoms. Our two test groups were both cognitively healthy as defined after detailed neuropsychiatric testing and were biochemically differentiated based on cutoff levels that are established in AD for CSF $A\beta_{42}$ and total tau biomarkers.

Previous reports have evaluated retinal thickness changes in preclinical AD using OCT. Few have examined CSF protein levels, although exclusively with respect to Aß, and have found little or no correlation between OCT findings and neurologic parameters [43, 68]. Notably, however, Tau burden has been shown to be more consistent with level of cognitive impairment in AD than cerebral Aß accumulation [44, 69]. The current article analyzes, for the first time, retinal morphology in relation to both CSF Aß and Tau CSF levels. Strengths of the current study include not only rigorous selection of pre-symptomatic participants, but also using a cutoff for CSF Aß /Tau ratios that was previously demonstrated by our laboratory to correctly predict 85% of participants with AD.

Limitations of this study are those that would be expected in an exploratory approach. The statistical models require validation, and the findings need to be replicated and applied to different populations to assess both neurological and ophthalmological disease, physiological, and cultural/ethnic specificity. Nevertheless, the present article represents the largest study of RNFL as measured by OCT in preclinical AD. Our investigation was also cross-sectional in nature. Proper characterization of disease natural history requires a long-term longitudinal follow-up study. Therefore, we intend to use these results as baseline values for future longitudinal studies. Further investigation is necessary to precisely elucidate the mechanism of retinal pathology observed in these cohorts. Sensitive retinal biomarkers may be advantageous given their cost and time efficiency in addition to their non-invasiveness and high level of accessibility [44]. Therefore, the purpose of our study was to investigate retinal nerve fiber thickness as a potential screening tool for individuals at risk for AD, thereby maximizing sensitivity. Our laboratory is in the process of deriving a balanced cut-off that maximizes sensitivity in addition to specificity. OCT measurements have the immediate potential to allow rapid data acquisition in studies of AD patients or those at risk for AD with the objective of building a large database of RNFL thickness in different AD patient populations.

## Conclusions

We report that RNFL thickness of cognitively healthy older individuals predicts CSF amyloid/tau, the gold standard biomarker of AD pathology, with 87% sensitivity in a cross-sectional study. Because OCT is widely available, we think this data may inspire a large community health opportunity to evaluate screening for individuals at risk for AD.

## Supporting information

**S1 Fig. AD OCT scatter plots for OD: Series 1 = CH-PAT; Series 2 = CH-NAT.**
(DOCX)

**S1 Table.**
(DOCX)

**S2 Table.**
(DOCX)

## Acknowledgments

We are grateful to the altruistic involvement of our study participants.

## Author Contributions

**Conceptualization:** Samuel Asanad, Fred N. Ross-Cisneros, Michael G. Harrington.

**Data curation:** Samuel Asanad.

**Formal analysis:** Janice M. Pogoda, Michael G. Harrington.

**Funding acquisition:** Berislav V. Zlokovic, Michael G. Harrington.

**Investigation:** Samuel Asanad, Michele Fantini, William Sultan, Marco Nassisi, Christian M. Felix, Jessica Wu, Rustum Karanjia, Fred N. Ross-Cisneros, Abhay P. Sagare, Michael G. Harrington.

**Methodology:** Samuel Asanad, Rustum Karanjia, Berislav V. Zlokovic, Helena C. Chui, Janice M. Pogoda, Xianghong Arakaki, Alfred N. Fonteh, Alfredo A. Sadun A. A., Michael G. Harrington.

**Project administration:** Michael G. Harrington.

**Writing – review & editing:** Samuel Asanad, Michele Fantini, William Sultan, Marco Nassisi, Christian M. Felix, Jessica Wu, Rustum Karanjia, Fred N. Ross-Cisneros, Abhay P. Sagare, Berislav V. Zlokovic, Helena C. Chui, Janice M. Pogoda, Xianghong Arakaki, Alfred N. Fonteh, Alfredo A. Sadun A. A., Michael G. Harrington.

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
