## [Decision Letter · Decision Letter 0]

17 Feb 2020

PONE-D-20-02170

Retinal nerve fiber layer thickness predicts CSF amyloid/tau before cognitive decline

PLOS ONE

Dear Dr. Harrington,

Thank you for submitting your manuscript to PLOS ONE. After careful consideration, we feel that it has merit but does not fully meet PLOS ONE’s publication criteria as it currently stands. Therefore, we invite you to submit a revised version of the manuscript that addresses the points raised during the review process.

Both reviewers indicated that your research is technically sound and that the data are analyzed properly. However, reviewer two requests more thorough discussion of why your results differ from those of previous published studies suggesting macular assessment as being more sensitive in detecting neuronal changes in both AD and MCI patients.

We would appreciate receiving your revised manuscript by Apr 02 2020 11:59PM. To enhance the reproducibility of your results, we recommend that if applicable you deposit your laboratory protocols in protocols.io, where a protocol can be assigned its own identifier (DOI) such that it can be cited independently in the future. For instructions see: http://journals.plos.org/plosone/s/submission-guidelines#loc-laboratory-protocols

We look forward to receiving your revised manuscript.

Kind regards,

Alfred S Lewin, Ph.D.

Academic Editor

PLOS ONE

Journal Requirements:

http://iovs.arvojournals.org/article.aspx?articleid=2731003

In your revision ensure you cite all your sources (including your own works), and quote or rephrase any duplicated text outside the methods section. Further consideration is dependent on these concerns being addressed.

3. Please upload a copy of Supporting Information Figures and Tables, which you refer to in your text on page 19-25.

Reviewers' comments:

Reviewer's Responses to Questions

**Comments to the Author**

1. Is the manuscript technically sound, and do the data support the conclusions?

Reviewer #1: Yes

Reviewer #2: Yes

2. Has the statistical analysis been performed appropriately and rigorously? 

Reviewer #1: Yes

Reviewer #2: Yes

3. Have the authors made all data underlying the findings in their manuscript fully available?

Reviewer #1: Yes

Reviewer #2: Yes

4. Is the manuscript presented in an intelligible fashion and written in standard English?

Reviewer #1: Yes

Reviewer #2: Yes

5. Review Comments to the Author

Reviewer #1: This is a well written paper where Harrington et al correlate cerebrospinal fluid biomarkers with retinal nerve fiber layer.

However, minor changes might improve the quality of the manuscript:

First, I cannot see the number of cases included in the abstract.

Second, the conclussion might be a very categorical statement: “ Retinal data have sufficient sensitivity to predict which patients have CSF biomarkers of AD pathology before cognitive deficits are detectable”, as this is only one study with a small number of patients.

In line 113, the abbreviation of RGC-IPL should be explained.

Some references have a point after the journals and other not. Please amend the references section.

Reviewer #2: The authors included cognitively healthy individuals and divided them in to groups, based on normal or pathological CSF Abeta42 and tau ratio. They correlated and compared the two gropus using OCT-SD RNFL, ganglion cell-inner plexiform layer and macular thickness. They found a RNFL thickness reduction in cognitively healthy individuals with pathological CSF. They concluded RNFL thickness can be used with sufficient sensitivity to predict wich patients have CSF biomarkers of AD pathology before cognitive are detectable.

The results are interesting pointing toward the valeu of OCT as a pre-clinical predictor in AD patients. However, the results seems to be somewhat conflicting with many previous publications that point to macular assessment as being more sensitive in detecting neuronal changes in both AD and MCI patients. How do the authors justify these differences? This should be addressed in more depth in the discussion. Studies involving patients with MCI show changes in macular thickness, especially with ganglion cell complexes around the central 3 mm of the fovea, while RNFL was preserved in these patients (Almeida ALM, Pires LA, Figueiredo EA, Costa-Cunha LVF, Zacharias LC, Preti RC, Monteiro MLR, Cunha LP. Correlation between cognitive impairment and retinal neural loss assessed by swept-source optical coherence tomography in patients

with mild cognitive impairment. Alzheimers Dement (Amst). 2019) In these studies, it was also demonstrated that the thickness analysis divided into 9 sectors (and not six as performed in the present study) according to the ETDRS map seems to be more sensitive. These similar findings were also demonstrated in patients with AD in other studies. (Santos CY, Johnson LN, Sinoff SE, Festa EK, Heindel WC, Snyder PJ. Change in retinal structural anatomy during the preclinical stage of Alzheimer's disease.Alzheimers Dement (Amst). 2018 Feb 7;10:196-209. Cunha LP, Lopes LC, Costa-Cunha LV, Costa CF, Pires LA, Almeida AL, Monteiro ML. Macular Thickness Measurements with Frequency Domain-OCT for Quantification of Retinal Neural Loss and its Correlation with Cognitive Impairment in Alzheimer's Disease. PLoS One. 2016 Apr 22;11(4):e0153830.

There is sufficient evidence suggesting that there is a primary involvement of the retina in patients with AD and that these changes (neuronal impairment with decreased ganglion cell complex) may occur earlier than axonal loss, which contrasts with the findings of the present study.

6. PLOS authors have the option to publish the peer review history of their article (what does this mean?). If published, this will include your full peer review and any attached files.

Reviewer #1: Yes: Alfonso Casado

Reviewer #2: No

---

## [Author Response · Author response to Decision Letter 0]

6 Apr 2020

For the initial journal requirements:

1. We have checked and complied with PLOS ONE’s publication criteria.

2. We have changed the overlapping text from a previous paper at the start of the Intro.

3. We have included the Supp Figures and tables.

Reviewer 1:

Comment 1:

“First, I cannot see the number of cases included in the abstract.”

Authors reply

We thank the reviewer for this comment and have clarified the number of cases in the abstract. 

Comment 2:

“Second, the conclusion might be a very categorical statement: “Retinal data have sufficient sensitivity to predict which patients have CSF biomarkers of AD pathology before cognitive deficits are detectable”, as this is only one study with a small number of patients.”

Authors reply

We appreciate this comment and agree that this may be viewed as a categorical statement. We modify the manuscript by qualifying our conclusion according to the findings specific to our study. 

Comment 3:

“In line 113, the abbreviation of RGC-IPL should be explained.”

Authors reply

We thank the reviewer for this comment and have modified the manuscript accordingly: 

Comment 4:

“Some references have a point after the journals and other not. Please amend the references section.”

Authors reply

We appreciate this comment and we have amended the References accordingly. 

Reviewer 2:

Comment 1:

“The results seems to be somewhat conflicting with many previous publications that point to macular assessment as being more sensitive in detecting neuronal changes in both AD and MCI patients. How do the authors justify these differences? This should be addressed in more depth in the discussion. Studies involving patients with MCI show changes in macular thickness, especially with ganglion cell complexes around the central 3 mm of the fovea, while RNFL was preserved in these patients (Almeida, et al. Alzheimers Dement 2019).” 

Authors reply

We appreciate this comment and agree with providing a more in-depth discussion of our findings relative to previous studies. In agreement with the reviewer, we acknowledge previous findings suggesting progressive retinal thinning from MCI to AD. As referenced by the reviewer, Almeida et al. found ganglion cell complex thinning. We would like to explain that retinal thickness findings in MCI have been controversial. We add to the manuscript by describing previous studies conducted by Ascaso et al. and Oktem et al., which showed significant RNFL thinning in MCI and this thinning was found to significantly correlate with Mini-Mental State Examination (MMSE) scores (Ascaso et al., 2014; Oktem et al., 2015). We also reference studies, which showed no significant retinal thinning in MCI (Pillai et al., 2016; Feke et al., 2015). We further revise the manuscript by describing potential confounding factors that reportedly contribute to conflicting findings between studies including variability in OCT modalities, diagnostic criteria for subject enrollment, as well as the heterogeneity of MCI. We have modified the manuscript Discussion and have included the corresponding References accordingly.

Comment 2:

“In these studies, it was also demonstrated that the thickness analysis divided into 9 sectors (and not six as performed in the present study) according to the ETDRS map seems to be more sensitive.” 

Authors reply

We appreciate and thank the reviewer for this comment. We would like to clarify that measurements for the GC-IPL were acquired using the standard 6-retinal sector elliptical annulus. With respect to macular full-thickness, measurements were determined using 9-retinal subfields, as defined by the Early Treatment Diabetic Retinopathy Study (ETDRS) as shown in the newly added, Figure 1C. This is consistent with previously published protocols using the spectral-domain OCT (Cirrus, Zeiss). As referenced by the reviewer, Almeida et al. and Cunha et al. similarly measured macular thickness using the ETDRS grid. We would like to mention that Santos et al. measured both ganglion cell complex macular full-thickness using a 4-sector grid. In addition, Almeida et al., Cunha et al., and Santos et al., each differ in their OCT technology and device models (SS-OCT, fd-OCT, and Heidelberg SPECTRALIS SD-OCT, respectively), which intrinsically vary in computational analysis. As suggested by the reviewer, we acknowledge that differences in OCT devices and segmentation algorithms may contribute to variable retinal thickness findings. We have modified the manuscript Methods and Discussion sections to reflect these changes.

Comment 3:

“These similar findings were also demonstrated in patients with AD in other studies (Santos CY, et al. Alzheimers Dement 2018 and Cunha LP, et al. PLoS One 2016). There is sufficient evidence suggesting that there is a primary involvement of the retina in patients with AD and that these changes (neuronal impairment with decreased ganglion cell complex) may occur earlier than axonal loss, which contrasts with the findings of the present study.”

Authors reply

We appreciate this comment and agree with further discussing our findings in relation to previous studies in AD. As suggested by the reviewer, we justify our findings by describing previous studies in AD, which have similarly shown RNFL thinning and a significant correlation to visual function and cognitive deficits43-52. We would like to mention that the referenced AD study conducted by Cunha et al. study showed both ganglion cell complex and RNFL axonal loss; and these were both shown to significantly correlate with cognitive function testing. In addition, studies conducted by Martin et al. on retinal thickness changes in relation to AD duration and severity showed RNFL thinning preceded ganglion cell loss. We further support our results by discussing recent findings reported by our laboratory in AD using postmortem histopathology54. We would also like to point out that Santos et al. primarily found RNFL axonal loss in preclinical AD57. Ganglion cell layer, inner plexiform layer, as well as outer nuclear layer thickness changes were regarded as non-significant findings. We have modified the manuscript Discussion and the corresponding References to reflect these changes.

We are grateful to the reviewers for their supportive and helpful comments and for giving us the opportunity to submit a revised and improved version of our manuscript. We have responded to all the points raised by the reviewers and yourself, modifying the manuscript accordingly. Our point-by-point answers are provided above and in the corrected manuscript. We hope that you now find the revised tracked and clean manuscript versions suitable for publication. Please let us know if there are further questions.

---

## [Decision Letter · Decision Letter 1]

22 Apr 2020

Retinal nerve fiber layer thickness predicts CSF amyloid/tau before cognitive decline

PONE-D-20-02170R1

Dear Dr. Harrington,

We are pleased to inform you that your manuscript has been judged scientifically suitable for publication and will be formally accepted for publication once it complies with all outstanding technical requirements.

With kind regards,

Alfred S Lewin, Ph.D.

Section Editor

PLOS ONE

Additional Editor Comments (optional):

Reviewers' comments:

Reviewer's Responses to Questions

**Comments to the Author**

1. If the authors have adequately addressed your comments raised in a previous round of review and you feel that this manuscript is now acceptable for publication, you may indicate that here to bypass the “Comments to the Author” section, enter your conflict of interest statement in the “Confidential to Editor” section, and submit your "Accept" recommendation.

Reviewer #1: All comments have been addressed

Reviewer #2: All comments have been addressed

2. Is the manuscript technically sound, and do the data support the conclusions?

Reviewer #1: Yes

Reviewer #2: Yes

3. Has the statistical analysis been performed appropriately and rigorously? 

Reviewer #1: Yes

Reviewer #2: Yes

4. Have the authors made all data underlying the findings in their manuscript fully available?

Reviewer #1: Yes

Reviewer #2: Yes

5. Is the manuscript presented in an intelligible fashion and written in standard English?

Reviewer #1: Yes

Reviewer #2: Yes

6. Review Comments to the Author

Reviewer #1: All suggested changes have been made and the manuscript have improved. The results are interesting and their relevance is medium-high.

Reviewer #2: (No Response)

7. PLOS authors have the option to publish the peer review history of their article (what does this mean?). If published, this will include your full peer review and any attached files.

Reviewer #1: Yes: Alfonso Casado, MD, PhD,Department of Ophthalmology, Hospital Universitario Marqués de Valdecilla, IDIVAL

Reviewer #2: Yes: Leonardo Provetti Cunha

---

## [Editor Report · Acceptance letter]

12 May 2020

PONE-D-20-02170R1 

Retinal nerve fiber layer thickness predicts CSF amyloid/tau before cognitive decline 

Dear Dr. Harrington:

I am pleased to inform you that your manuscript has been deemed suitable for publication in PLOS ONE. Congratulations! Your manuscript is now with our production department. 

With kind regards,

on behalf of

Dr. Alfred S Lewin 

Section Editor

PLOS ONE